# SiO_2_ Passivated Graphene Saturable Absorber Mirrors for Ultrashort Pulse Generation

**DOI:** 10.3390/nano13010111

**Published:** 2022-12-26

**Authors:** Hongpei Wang, Cheng Jiang, Huiyuan Chu, Hao Dai, Beibei Fu, Shulong Lu, Ziyang Zhang

**Affiliations:** 1School of Nano-Tech and Nano-Bionics, University of Science and Technology of China, Hefei 230026, China; 2School of Electronic and Information Engineering, Qingdao University, Qingdao 266071, China

**Keywords:** ultrashort pulse fiber laser, saturable absorber, graphene, single-layer graphene

## Abstract

Owing to its broadband absorption, ultrafast recovery time, and excellent saturable absorption feature, graphene has been recognized as one of the best candidates as a high-performance saturable absorber (SA). However, the low absorption efficiency and reduced modulation depth severely limit the application of graphene-based SA in ultrafast fiber lasers. In this paper, a single-layer graphene saturable absorber mirror (SG-SAM) was coated by a quarter-wave SiO_2_ passivated layer, and a significantly enhanced modulation depth and reduced saturation intensity were obtained simultaneously compared to the SG-SAM without the SiO_2_ coating layer. In addition, long-term operational stability was found in the device due to the excellent isolation and protection of the graphene absorption layer from the external environment by the SiO_2_ layer. The high performance of the SAM was further confirmed by the construction of a ring-cavity EDF laser generating mode-locked pulses with a central wavelength of 1563.7 nm, a repetition rate of 34.17 MHz, and a pulse width of 830 fs.

## 1. Introduction

Ultrashort pulse fiber lasers have attracted tremendous attention due to their excellent flexibility, compactness, stability, efficient heat dissipation, high beam quality, and convenience in coupling the output to fiber systems [1,2,3,4,5,6,7]. Among those ultrashort pulse lasers, the passively mode-locked fiber laser based on a saturable absorber (SA) is the most powerful strategy to develop ultrashort pulses because of the benefits of their simple structure, high beam quality, alignment-free compact design, and excellent compatibility [8,9,10,11,12,13]. Since graphene was first suggested in 2009 as an optical SA for mode-locked ultrafast fiber lasers [14], the graphene SA (G-SA) has exhibited outstanding performance, expediting huge prospects for SA in the research field of ultrafast fiber lasers. The unique characteristics of graphene, such as its ultra-wide operating bandwidth, ultrafast recovery time, low saturable absorption, great relative modulation depth, and wavelength-independent operation, make it memorable and allow it to perform efficiently as an SA to build wideband mode-locked ultrafast fiber laser pulses [15,16,17,18]. G-SA has been employed in pulsed lasers to generate ultrashort pulses [19]. The linear dispersion of Dirac electrons intrinsic to single-layer graphene (SLG) revealed its bandwidth-independent zero-bandgap semiconductor nature, which makes it an important SA for 1, 1.5, and 2 μm mode-locked fiber lasers in the near-infrared (NIR) region [20,21,22]. However, the small modulation depth by the ~2.3% low absorption efficiency per graphene layer significantly weakens the application of the G-SA in the research area of ultrafast lasers [23,24,25]. Stacking multiple graphene layers could increase the optical absorption ability, but the defects induced in the interfaces between each graphene layer will result in the deteriorated performance of the device. Transfer of SLG film onto the surface of D-shaped and tapered fibers to interact with the evanescent field can enhance the light-graphene interaction while reducing the thermal damage of graphene [26,27,28,29,30], which is beneficial for the long-term operational stability of G-SA. However, this approach inevitably sacrifices the integrity of the fiber structure and transmission modes, limiting their development space and potential. The quarter-wave SiO_2_ enhancement layer was proved to optimize the negative group-delay dispersion (GDD) and nonlinear modulation of SESAM to improve its modulation depth [31,32,33]. In addition, SiO_2_ is also used as a protective layer on the surface of SAs materials to isolate the effects of the external environment, increase the damage threshold, and improve the long-term stability of the device [34,35,36,37].

In this study, SLG saturable absorption mirrors (SAM) were prepared by PMMA method [38,39] transfer on Au mirrors. Then, a quarter wave of SiO_2_ was deposited as the enhancement layer by plasma enhanced chemical vapor deposition (PECVD) to obtain SG-SAM. It has been demonstrated that the SiO_2_ enhancement layer can increase the modulation depth, and also can decrease the saturation intensity of SG-SAM as well. At the same time, the enhancement layer covers the graphene surface, isolating it from the external environment and improving the device’s long-term operational stability. A comparison of the Q-switched output properties in a home-built EDF fiber ring cavity shows the obviously enhanced output performance of the SG-SAM device. A stable mode-locked pulsed laser with a central wavelength of 1563.7 nm, a repetition rate of 34.17 MHz, and a pulse width of 830 fs were realized. This work shows a promising method for the fabrication of high-performance graphene-based SAM and could be extended to the application of graphene-based passive devices in the area of optoelectronics.

## 2. Sample Preparation and Characterization

A graphene saturable absorbent mirror (G-SAM) was fabricated using SLG grown on Cu foils (30 µm thick Alfa-Aesar, Shanghai, China, purity 99.99%) via chemical vapor deposition (CVD). The preparation procedure is illustrated in Figure 1a. First, an Au film with a thickness of 500 nm was deposited on the surface of the Si substrate via magnetron sputtering as a reflection structure for the SAM. And then, the SLG was uniformly transferred from the Cu surface to the Au/Si reflection structure using the PMMA method to obtain the G-SAM. Finally, a quarter-wave SiO_2_ as an enhancement layer was deposited on the surface of G-SAM via PECVD facility to get SG-SAM for improving the nonlinear optical properties, such as modulation depth ∆*R*, saturation intensity *I*_s_, and non-saturable losses *R*_ns_ of the G-SAM. In this work, the deposition parameters were: radio frequency (RF) power of 20 W, temperature of 350 °C, chamber pressure of 2000 mTorr, and the flow rates of SiH_4_, N_2_O, and N_2_ are 4, 710, and 180 sccm, respectively. The thickness of the capping layer after 4 min deposition was measured by the ellipsometer (iSE J.A. Woollam Co., Inc., Lincoln, NE, USA) as 270 nm. Moreover, the SiO_2_ coating on the surface insulates the graphene from the external environment’s effects, which helps improve the device’s long-term stability. Typical optical microscopy (OM) of G-SAM is shown in Figure 1b, displaying a clear boundary between the transferred graphene and the gold film, which confirms the graphene is successfully transferred. The SG-SAM can uniformly cover the Au/Si surface with an area of 5 × 5 mm^2^, as shown in Figure 1c. Figure 1d shows the corresponding enlarged images taken from random positions of the substrate, in which the highly flat and homogeneous surface is successfully obtained, confirming the high quality of SG-SAM in the whole area.

The energy dispersive spectroscopy (EDS) spectrum was carried out to further investigate SG-SAM’s characterization, as shown in Figure 2a. The EDS spectrum exhibits compositions containing carbon, silicon, and oxygen, which confirms the graphene has been successfully transferred onto the Au/Si reflection structure. Meanwhile, the inset image in Figure 2a shows an SEM picture of the SG-SAM. It can be seen that the highly flat and homogeneous morphology of the SG-SAM further verifies its high quality. The C, O, Si, and Au elements contained in the SG-SAM were shown in the element mappings in Figure 2b. The uniform distribution of the C element indicates that the transfer of SLG is highly uniform, and no large voids are observed. Furthermore, the quality and number of layers are investigated by Raman spectrum spectroscopy. The Raman spectrum obtained for SG-SAM is exhibited in Figure 2c. The SAM shows two clear Raman peaks at 1584 and 2675 cm^−1^ corresponding to the G band and 2D band of graphene [40,41], respectively. The peak intensity of the 2D band is considerably higher than that of the G band, confirming the presence of only one graphene layer in the SAM. Moreover, the D peak originates from the effect caused by defects, which can reflect the degree of defects. In this work, the D peak is almost negligible, indicating that the SAM prepared is a single layer with good crystallinity. After the deposition of the SiO_2_ layer, the absorption spectrum of the SG-SAM was measured by using a broadband light source, as shown in Figure 2d. In this spectrum, the SG-SAM illustrates a broad wavelength absorption spectrum in the 1000 to 1800 nm band, indicating a broadband response characteristic of the NIR region. The sample has an absorption of 9.73% in the 1550 nm band.

As shown in Figure 3a, a balanced twin-detector nonlinear saturable adsorption test system was constructed using a self-made mode-locked fiber laser (center wavelength of 1560 nm, repetition rate of 11.4 MHz, and pulse width of 820 fs) in the erbium-doped fiber amplifier (EDFA), and a 50:50 optical coupler (OC) output section to connect power meter 1, the test SAM, and power meter 2. The nonlinear saturation absorption curves of G-SAM and SG-SAM are shown in Figure 3b and 3c, respectively. After fitting by the following equation [42]: *R* = 1 − Δ*R* × exp(−*I/I*_s_) − *R*_n_, where *R* is the reflectivity, ∆*R* is the modulation depth, *I* is the input intensity, *I*_s_ is the saturation intensity, and *R*_ns_ is the non-saturable losses, respectively. The measured *I*_s_, ∆*R*, and *R*_ns_ are 0.41 MW/cm^2^, 2.8%, and 18.8% for G-SAM and 0.35 MW/cm^2^, 6.8%, and 22.6% for SG-SAM, respectively. The decrease of *I*_s_ and the increase of Δ*R* after deposition is attributed to the increase of the surface field intensity enhancement factor of the device by the quarter-wave layer SiO_2_ (low-index material) [31]. The increase of *R*_ns_ may attribute to the not perfectly densed SiO_2_ layer structure by PECVD method. The finite-difference time-domain (FDTD) method was used to simulate the electric field distribution of G-SAM and SG-SAM to further investigate the effect of the SiO_2_ enhancement layer on the field intensity of the SAM surface, as shown in Figure 3d,e. It shows that the SLG is located at the zero point in the Z-direction, below that is Au, and above that is SiO_2_/Air. The wavelength of the irradiation source is at 1.55 μm. Due to the existence of the enhancement layer, the SG-SAM shows a high photon absorption feature at 1.55 μm range, which is in alignment with the much stronger surface electric field strength in SG-SAM compared to that of G-SAM.

## 3. Results and Discussion

As shown in Figure 4a, an EDF ring cavity laser was constructed to investigate the SAMs’ Q-switching and mode-locking characteristics. The ring cavity fiber lasers mainly consist of the following components such as 980 nm laser diode (LD) with a maximum output power of 500 mW, 980/1550 nm wavelength division multiplexer (WDM), 1.5 m EDF, polarization independent isolator (PI-ISO), polarization controller (PC), circulator (CIR), SAM and 10/90 OC. All components are contacted with single-mode fiber. Among them, the 980 nm LD serves as the pump source to provide the pump light for the ring cavity fiber laser and is coupled to the fiber laser using WDM; PI-ISO is used to prevent laser reflection and to ensure a non-directional propagation of light in the cavity; the PC can adjust the polarization state and birefringence to assist the Q-switching and mode-locking; the output coupler (OC) provides 10% of the laser output light in the ring cavity for analysis, and the rest is transmitted back to the cavity via the WDM. A digital storage oscilloscope (DSOS054A Keysight Inc., Santa Rosa, CA, USA), autocorrelator (FR-103XL Femtochrome Inc., Berkeley, CA, USA), optical spectrum analyzer (MS9740A Anritsu Co., Ltd., Kanagawa, Japan), spectrum analyzer (N9322C Keysight Inc., Santa Rosa, CA, USA), and digital display power meter was used to analyze the pulse train, pulse profile, optical spectrum, RF spectrum, and output power of the output laser.

The passive Q-switched lasers were constructed using G-SAM and SG-SAM in designed ring cavities to investigate the laser performance. The pump power reaches 22 mW and 26 mW, corresponding to the continuous Q-switched modes of G-SAM and SG-SAM, respectively. Figure 4b,c show the G-SAM and SG-SAM Q-switched pulse trains at different pump powers, respectively. Evidently, during the tuning process of pumping power, the output pulse train remains stable and relatively homogeneous, the intensity distribution is reasonable, and the repetition frequency continuously increases. It indicates that the fiber laser operates in a highly stable Q-switched state. When the pump power is increased beyond 122 mW and 340 mW, respectively, the impulsive state suddenly disappears, a continuous wave (CW) state is obtained, and the Q-switched state ends, which clearly shows after deposition enhancement layer, SG-SAM can achieve stable Q-switching output at higher pump power than G-SAM. Figure 4d,e show the G-SAM and SG-SAM optical spectra, in which the corresponding central wavelength was found to be 1559 nm and 1558 nm, respectively.

Figure 5 demonstrates the evolution of output power, repetition frequency, duration time, and single pulse energy as a function of pump power. It exhibits typical features of Q-switched mode. As shown in Figure 5a,b, the output power of G-SAM and SG-SAM linearly increases from 0.09 mW to 0.96 mW and 0.08 mW to 2.83 mW, respectively, as the pump power increases. The repetition frequency linearly increases from 12.55 kHz to 41.2 kHz and 12.32 kHz to 79.25 kHz, respectively. Figure 5c,d demonstrate the pulse duration time varies from 16.63 μs to 3.85 μs and 12 μs to 2.45 μs, respectively, with the increase of pump power, and the single pulse energy varies between 6.96–23.4 nJ and 6.63–35.73 nJ, respectively. At the low pump power range, the pulse duration time decreases rapidly, presumably due to the rapid accumulation of electrons at higher energy levels. However, due to supersaturation, the accumulation rate slows down, resulting in small duration time changes at high pump power ranges. From the obtained results shown in Figure 5a–d, it is clear that the SG-SAM with enhancement layer has a higher output power, higher repetition frequency, narrower duration time, and stronger single pulse energy than that of the G-SAM.

The SG-SAM was inserted into the designed ring cavity to construct a passive mode-locking laser. The total cavity length was changed to 6.0 m to adjust for dispersion and loss in the cavity [43,44]. Self-starting mode-locking occurs with a threshold pump power of 29 mW. The mode-locked state can be maintained up to the maximum pump power of 500 mW. Figure 6a illustrates the mode-locked pulse train at the threshold, which is very homogeneous and stable. It shows a pulse period of 29.3 ns, corresponding to a calculated repetition frequency of 34.17 MHz. The single-to-noise ratio (SNR)of the RF spectrum is 50 dB, as shown in Figure 6b. To illustrate the high stability of the mode-locked laser, the RF spectrum in the 200 MHz range was accumulated as shown in Figure 6c, exhibiting a ~34.17 MHz repetition frequency, in which the pulse fundamental frequency is in agreement with the laser cavity length in the experiment. Figure 6d shows the mode-locked pulse autocorrelation trace with a measured full width at half maximum (FWHM) of 1.28 ps. The output pulses were fitted with the sech^2^ function, and the pulse width is about 830 fs from the autocorrelation trace. As shown in Figure 6e, the optical spectrum has a central wavelength of 1563.7 nm and a 3 dB bandwidth of 3.33 nm. The time-bandwidth product (TBP) was calculated to be 0.339, which indicates that the output pulse was slightly chirped. Figure 6f records the output power of the mode-locked laser as a function of the pump power, which increases linearly. The slope of the fitted line is 2.2%. When the self-starting mode-locking operation is switched on, the laser can operate at maximum pump power without further adjusting the PC as the pump power gradually increases.

The optical spectrum and output power of the laser at a pump power of 470 mW were recorded for 60 h, which confirms the long-term stability of the mode-locked laser operation. As shown in Figure 7a, neither the central wavelength drift nor spectral intensity fluctuations were observed during the measurement, which further indicates the long-term stable mode-locking operation. Meanwhile, the recorded output power of the mode-locked laser also remains stable, as shown in Figure 7b. These demonstrate the excellent long-term stability and higher output power of the SG-SAM mode-locked fiber laser.

## 4. Conclusions

To summarize, a quarter-wave SiO_2_ as an enhancement layer could be utilized for the fabrication of a high-quality SG-SAM, by which a large modulation depth and long-term operating stability could be realized simultaneously. Compared to the G-SAM, the SG-SAM exhibits better output characteristics and can operate at a considerably higher pump power level. In addition, a ring-cavity EDF laser based on the SG-SAM has been constructed, exhibiting superior passive mode-locked pulse operation with a pulse width of 830 fs. This work shows a potential way for the application of graphene-based passive devices in the area of optoelectronics.

## Figures and Tables

**Figure 1 nanomaterials-13-00111-f001:**
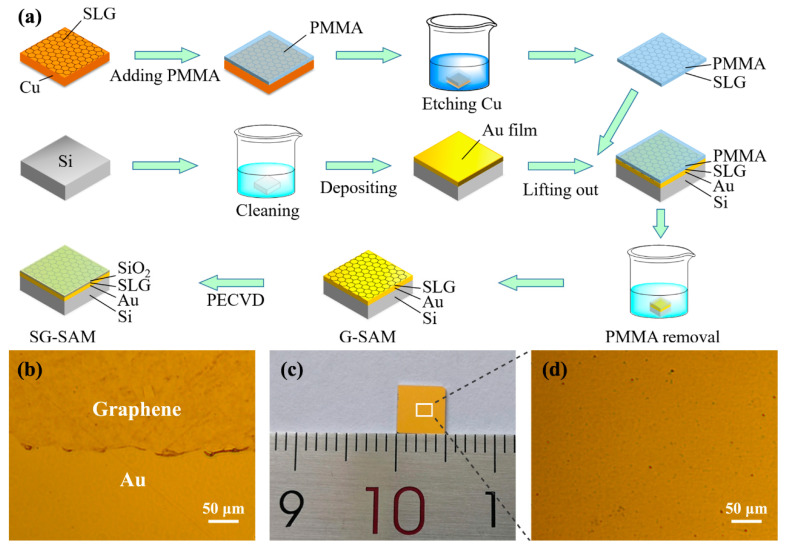
(**a**) Schematic illustration of G-SAM and SG-SAM preparation process: (**b**) optical microscopy images of G-SAM; (**c**) Photograph of the SG-SAM with the size of around 0.5 × 0.5 cm^2^.; and (**d**) optical microscopy images of SG-SAM.

**Figure 2 nanomaterials-13-00111-f002:**
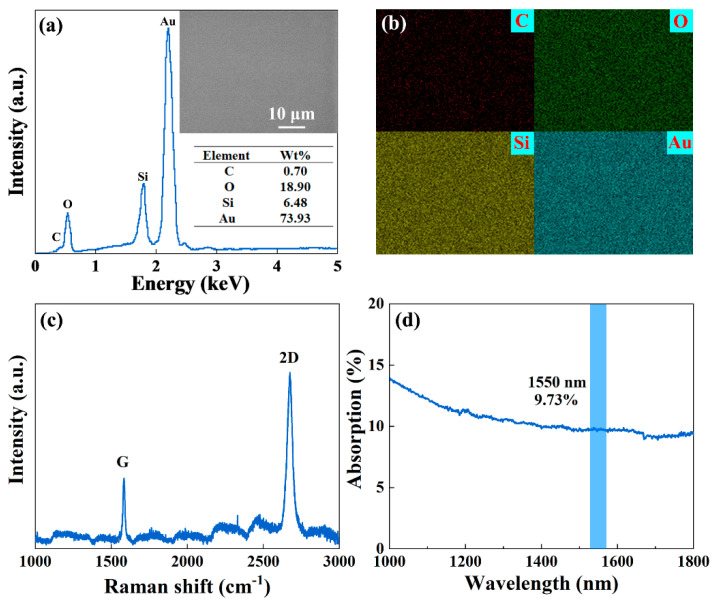
Characterizations of SG-SAM: (**a**) EDS spectrum, inset image is a SEM picture of the SG-SAM; (**b**) EDS element mappings; (**c**) Raman spectrum; and (**d**) absorption optical spectrum.

**Figure 3 nanomaterials-13-00111-f003:**
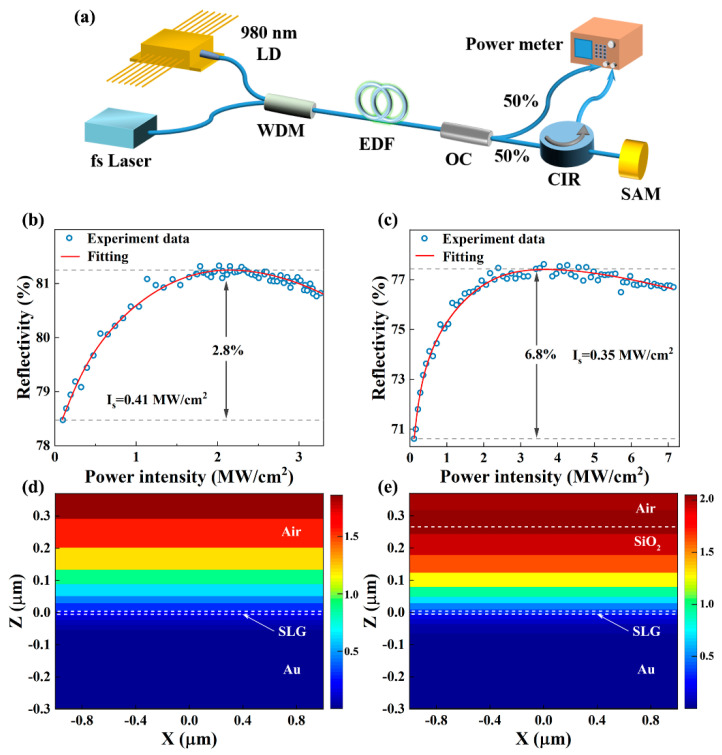
(**a**) Schematic illustration of the balanced twin-detector measurement. Nonlinear saturable absorption curves of G-SAM (**b**); and SG-SAM (**c**); electric field distribution of G-SAM (**d**); and SG-SAM (**e**).

**Figure 4 nanomaterials-13-00111-f004:**
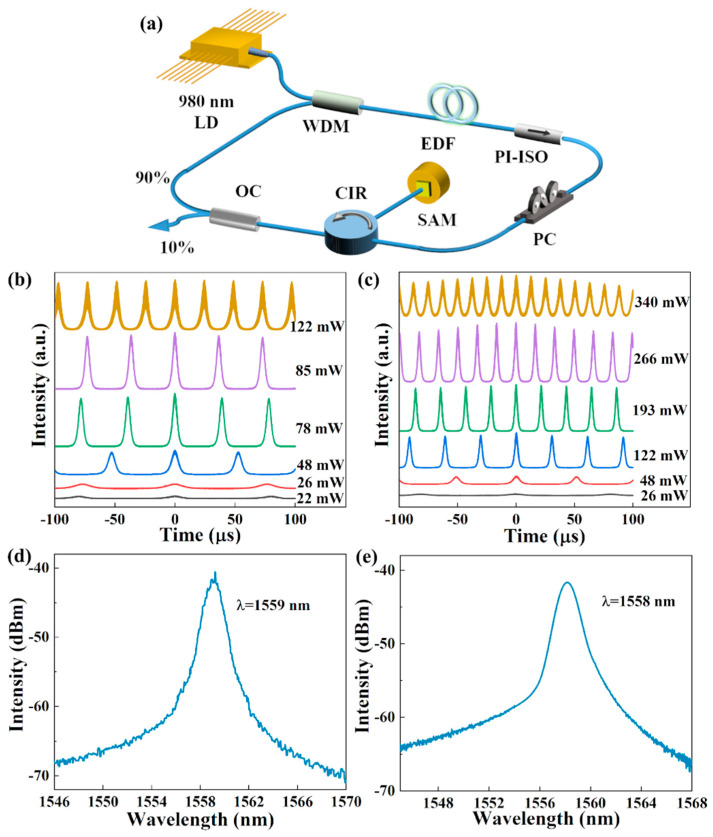
Output characteristics of the passive Q-switched laser: (**a**) schematic diagram of the experimental setup. Corresponding pulse train of G-SAM (**b**); and SG-SAM (**c**) evolution under different pump powers. Corresponding optical spectrum of G-SAM (**d**); and SG-SAM (**e**).

**Figure 5 nanomaterials-13-00111-f005:**
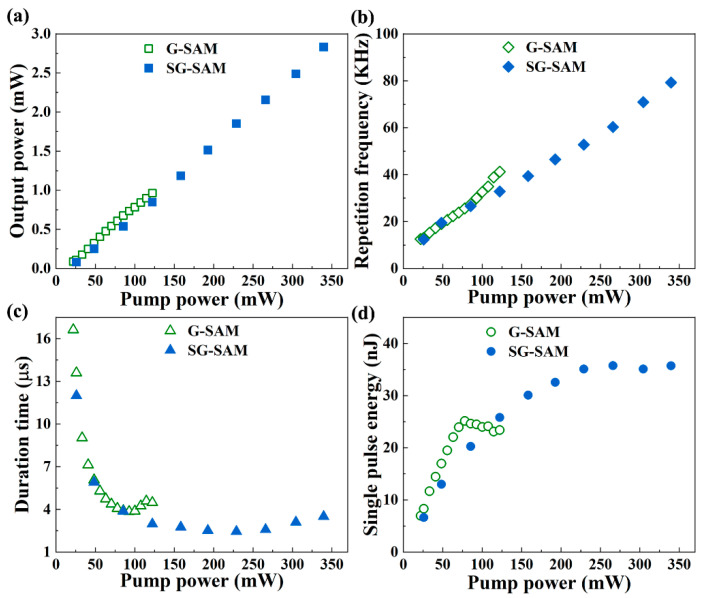
Q-switched pulse output characteristics. The output power (**a**); repetition frequency (**b**); duration time (**c**); and single pulse energy (**d**) evolution versus the pump power.

**Figure 6 nanomaterials-13-00111-f006:**
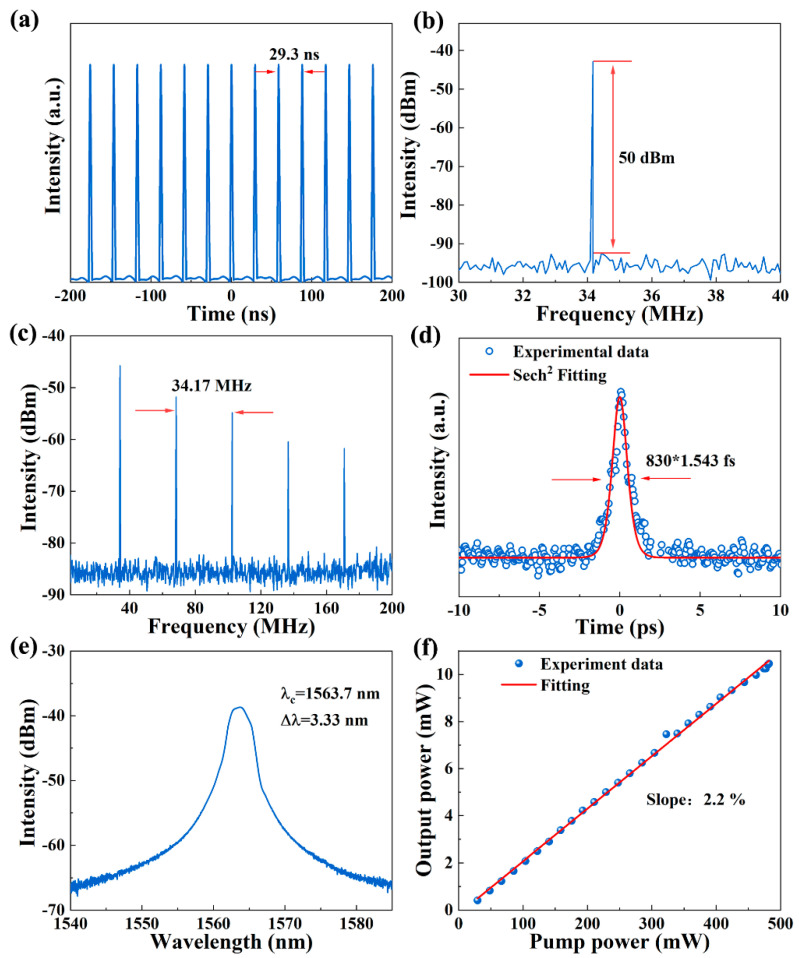
Characterization of mode-locked EDF fiber laser based on SG-SAM: (**a**) mode-lock pulse train; (**b**) RF spectrum at the fundamental frequency; (**c**) wideband RF spectrum in the range of 200 MHz; (**d**) autocorrelation trace; (**e**) ptical spectrum; and (**f**) output power as a function of pump power.

**Figure 7 nanomaterials-13-00111-f007:**
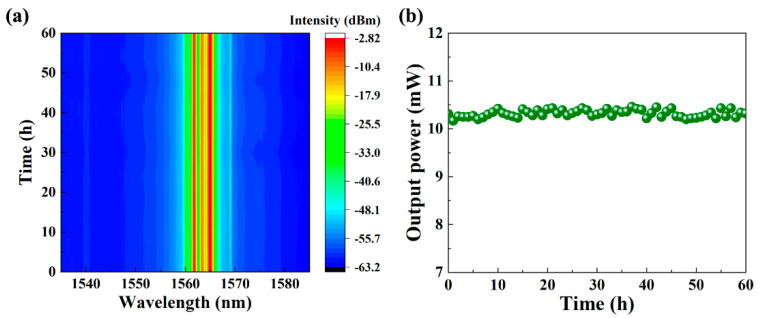
Long-term stability measurement of the mode-locked laser: (**a**) time stability of wavelength; and (**b**) time stability of output power.

## Data Availability

Not applicable.

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
