# Peer review of "SiO2 Passivated Graphene Saturable Absorber Mirrors for Ultrashort Pulse Generation"

_nanomaterials, 2022, doi:10.3390/nano13010111_

Round 1

Reviewer 1 Report

The manuscript entitled “SiO2 passivated graphene saturable absorber mirrors for ultra-2 short pulse generation” by Hongpei Wang et al. reports on experiments for the realisation of graphene saturable absorber enriched with a layer of SiO2.

The authors describe the fabrication method, the optical characterisation by doing spectroscopy and they demonstrate its usage for Q-switching.

Graphene based passive optical components are nowadays part of novel technology in photonics and the authors  show specifically how the presence of SiO2 coating layers improve the device performance.

The data are well presented, the strategy is well exposed and the main results (figs.5-6-7) are convincing.

I think the manuscript can be published in Nanomaterials.

Author Response

Dear  reviewer(s),

      We are very grateful for your comments on the manuscript.

Reviewer 2 Report

The authors have demonstrated the fabrication of the graphene-based saturable absorption mirror (GAM) and the usage of the GAM as a saturable absorbers (SAs) for ultrafast pulse generation at 1550 nm wavelength region. But, the graphene-based SAs in pulsed fibre lasers have been reported for many times. Besides, the measured pulse width (~830 fs) is not shortest with a graphene SA in mode-locked Er-doped fibre lasers that has been reported. I do not find any innovation or advantages that suitable for published. So, I recommend that the manuscript is not suitable for publication in this journal.

Author Response

Dear reviewer(s),

Thanks a lot for your comments. We agree that graphene-based SAs have been extensively investigated in the research area of ultrafast photonics. But there are some issues still remaining, such as low modulation depth, low damage threshold, and long-term operation instability that severely restrict the application of graphene-based devices. In this paper, we demonstrated that:

                   a quarter-wave SiO2 layer could be utilized for the fabrication of a high-quality SA, by which a large modulation depth (increased from the initial 2.8% to 6.8%) and long-term operating stability (constantly operate over 60hrs) could be realized simultaneously. And compared to the graphene-based SA without SiO2 enhancement layer, it exhibits better output characteristics and can operate at a considerably higher pump power level. In addition, the main fabrication technology utilized in this paper are general semiconductor fabrication process, and considering the gradually mature manufacturing level of large-scale graphene film, this work has great potential to expedite the applications for graphene-based passive devices in the area of optoelectronics.

Reviewer 3 Report

The manuscript lacks important discussions on laser operation and fundamental physics and engineering reasons behind it. It is rather listing and stating results of measurements. For example, I would like authors to elaborate more on the obtained results for mode-locked laser operation

1. What is the fitting curve shape shown for AC on Fig 6d. Gaussian? sech^2?

2. Nature of the pulse chirp? can the pulse be compressed externally? Any route/way to obtain shorter pulses?

3. Can authors reconfigure their autocorrelation and produce interferometric autocorrelation?

4. What is the reason for low out put power (~10 mW) a low efficiencies (~few percent)?

5. Reasons for stable mode-locking not being observed at Pp>500 mW

6. Any ideas on power scaling, e.g. larger MFD fiber , etc?

Round 2

Reviewer 2 Report

Accept

Reviewer 3 Report

The manuscript has been improved and reads better